# Contactless Detection of pH Change in a Liquid Analyte

**DOI:** 10.3390/s25092832

**Published:** 2025-04-30

**Authors:** Dylan Gustafson, Dominic Klyve

**Affiliations:** 1Department of Mathematics, Central Washington University, Ellensburg, WA 98926, USA; 2Know Labs, Inc., Seattle, WA 98104, USA; dominic@knowlabs.co

**Keywords:** sensor, pH, radiofrequency, contactless, microwave, non-invasive

## Abstract

We describe an experiment in which we employ a radiofrequency sensor to measure pH changes in a liquid solution. The experiment is novel in a few ways. First, the sensor does not have contact with the liquid but rather detects the change from the outside of a PVC pipe. Second, the change is detected using a Linear Discriminant Analysis model using values from an inverse Fourier transform of the frequency data as its features. We believe this to be the first use of Fourier analysis in contactless pH measurement using radio frequencies.

## 1. Introduction

Accurate and reliable measurement of pH levels is vital across a broad range of applications, including industrial processes [1], agricultural [2] and environmental monitoring [3,4], and biomedical diagnostics [5,6]. Conventional methods, such as optical and electrode-based pH meters, are widely used due to their demonstrated accuracy and ease of use. However, these methods have limitations, primarily stemming from the need for direct contact with the analyte.

Contactless sensing techniques have recently emerged as promising alternatives for non-invasive chemical analysis [7,8,9,10,11,12,13]. A sensor employing contactless measurement would be particularly suited for industrial applications where a shutdown required to repair or make changes to inline sensors would be too costly or where the analyte itself is too corrosive to put in direct contact with a sensor. Some earlier work has shown that RF may be a promising tool for the contactless detection of pH. One previous study demonstrated that pH can be measured using RF combined with chemical exchange saturation transfer (CEST) Magnetic Resonance Imaging (MRI) techniques [7]. While promising, this technique has the obvious limitation of requiring MRI hardware. Other work has attempted to differentiate between five standard pH buffer solutions by placing the solution directly on an RF planar microstrip antenna [8]. While promising, this work is limited by the use of a single resonant frequency in the analysis and by the fact that these five standard solutions are chemically different in ways other than their pH value.

RF-based pH sensors face significant challenges. Chief among these is its inability to pass through metal, the sensitivity required to measure minuscule changes in hydronium ion concentration, and the ability to measure pH consistently regardless of acid type or other substances present in solution. Just as traditional electrode-based pH meters are incompatible with particular substances such as organic solvents, hot caustics, soiling media, or abrasives [1], depending on their specific design, certain substances in a solution may obscure or overwhelm RF-based pH measurements, rendering them incompatible for a pH sensor of this kind. Addressing these challenges is crucial to advancing RF sensing technologies and expanding their applicability to real-world scenarios.

This study presents a novel approach to pH detection using a microwave-based sensor system that captures frequency response curves across a range of pH values. By applying discrete Fourier transform (DFT) analysis, we identify sinusoidal patterns in the frequency response data that correspond strongly with pH levels. These patterns enable the development of a simple yet robust algorithm for categorizing pH values, achieving high accuracy in our experimental tests over selected pH ranges. We believe this to be the first use of a DFT to analyze pH changes using RF spectroscopy.

## 2. Materials and Methods

The sensor employed in this study is the patented Know Labs, Inc. RF dielectric sensor, which consists of a printed circuit board assembly (PCBA) that generates RF signals and measures received power after passing those signals through an antenna array on a separate PCBA. The device measures the total received RF power at port 2 while transmitting a frequency sweep from port 1, which is a proxy for the forward transmission coefficient of the system, S_21_, digitized to a resolution of $10^{-5}$ volts. See [14] for a detailed description of the sensor assembly.

The analyte solution is contained in a 3-inch PVC pipe end cap (Part# 3P06 from NDS Inc., Woodland Hills, CA, USA), meant to mimic a cross-section of plastic pipe. This has a capacity of around 200 mL. The PCBA containing the sensor’s antenna array is held in place against the round wall of the PVC cap using a custom 3D-printed fixture designed to hold the board as close to the analyte as possible. The antenna board is connected to the RF-generating PCBA via SMP cables, which are controlled by a computer connected via USB. All components are taped down to the work surface to prevent them from moving during an experiment; however, the position of the antenna relative to the analyte is maintained much more precisely by fasteners and press-fit hardware. This hardware can be seen in Figure 1.

Kroger-brand distilled water was used as the baseline analyte. Due to its absorption of carbon dioxide from the air and lack of dissolved minerals to provide buffering, this water has a pH of around 5.5. During the experiment, the pH was lowered to a final “target” value by replacing some volume of the water with an equivalent volume of either 1 molar (M) sulfuric acid (H_2_SO_4_) or 1 M hydrochloric acid (HCl). For this study, the scope was limited to pH target values 1, 2, 3, 4, and 5. This meant that all final pH values could be achieved by adding acid to distilled water, as all of these target values were below 5.5. The goal of measuring only acidic pH values in this initial study was to characterize the sensor’s response specifically to the presence of hydronium ions, as opposed to hydroxide ions. Whole numbers were chosen simply out of convenience.

The procedure for each test run is as follows:Rinse out the PVC cap with distilled water and dry with a lint-free towel.Install the PVC cap and antenna board into the fixture.Measure out 200.0 g of distilled water on an AWS scale; part number AWS-600-BLK (American Weigh Scales, Cumming, GA, USA).Transfer the water into the cap, and allow it to settle for at least 20 s. Begin running microwave sweeps from 300 MHz to 4100 MHz at frequency intervals of 5 MHz.This first “baseline” run performs 300–4100 MHz sweeps for 120 s, generating 47 rows of data, before pausing.Once paused, remove the determined replacement volume of water using a micropipette.Add the same replacement volume of acid solution to the cap vessel using a micropipette.Mix in the acid using back-and-forth stirring motions to avoid creating a lasting vortex. Take care not to move any component of the test fixture; avoid bumping the walls of the cap vessel with the stirring stick.After stirring, wait at least another 30 s and then restart the sweeps.This final “change” run performs 300–4100 MHz sweeps for another 120 s, generating another 47 rows of data.Once the sweeps are complete and the script exits, dump the solution and rinse out the vessel with distilled water.

This procedure was performed with both HCl and H_2_SO_4_, targeting final pH values 1, 2, 3, 4, and 5, for a total of 10 different experiment types. Each experiment type was repeated 10 times (with the exception of the pH 1 HCl test, which was only performed 5 times), yielding a total of 95 runs of this procedure. For testing 200 mL of solution at a time, the acid replacement volumes needed to achieve each target pH value are given in Table 1. The H_2_SO_4_ replacement volumes were calculated using weak acid equilibrium calculations, assuming a pKa of 1.988 for bisulfate.

Additionally, 10 runs of a “mock” change experiment were performed for a grand total of 105 experiments. This mock experiment followed the same procedure as above, except none of the distilled water was replaced. During the pause between the two 47-sweep runs, the water was merely stirred in the same fashion.

## 3. Results

The device records each sweep result as a vector of voltage values. Each value represents the voltage measured in the receiving antenna when the transmitting antenna was generating a specific frequency. Since each sweep covers the frequency range 300 MHz to 4100 MHz, stepping upwards by 5 MHz, each sweep generates 760 voltage measurements. Plotting these data with voltage (V) on the vertical axis and frequency (MHz) on the horizontal axis creates a “frequency response curve”. Each of the 105 total experiments generated two sets of 47 frequency response curves to consider: “baseline” response curves for the distilled water and “final” response curves for the acidic solution.

Each experiment was first analyzed for signal drift by looking at the voltage values across its 47 “baseline” sweeps and 47 “final” sweeps. In the“baseline” series, the mean voltage values among the first 20 sweeps were compared with those of the remaining 27 sweeps. No statistically significant drift was found, which helped rule out the potential for false positives in a real-time measurement environment. Experiments were then summarized by taking the mean “final” curve minus the mean “baseline” curve, creating a mean “change” curve for each experiment. The mean “change” curves from the “mock” change experiments were used as a control against which to judge the “change” curves from the “real” experiments. Any feature that appeared in both the “real” change curves and the “mock” change curves was ignored.

Plotting the change curves revealed interesting sinusoidal behavior in the 1140–1890 MHz domain (Figure 2), while data outside this region did not appear useful. The “sine waves” in this limited region all have a wavelength of around 185 MHz and are in roughly the same phase, while their amplitudes increase with lower pH values (higher hydronium concentration). Interestingly, the amplitudes for the pH 2 HCl tests roughly match that of the pH 2 H_2_SO_4_ tests. Similarly, the pH 3 HCl amplitudes roughly match those of the pH 3 H_2_SO_4_ solutions. This is true for the pH 1 tests to some extent, although these response curves were noisier. All of this supports the idea that the sinusoidal shapes in the “change” curves are caused specifically by the hydronium ions in the solution, as opposed to the chloride or sulfate anions. This distinction is crucial for true pH measurement.

The goal of these data is to train a machine-learning model to predict pH from RF data. Accordingly, these 105 change curves were divided into a “training” set to train the model and a “test” set to evaluate its performance. With ten runs for each experiment type, the first five were placed in the training set, while the remaining five were placed in the test set. However, since the pH 1 HCl experiments had only had five runs in total, the first two of these runs went to the “training” set while the remaining three went to the “test” set, resulting in a total of fifty-two “training” experiments and fifty-three “test” experiments. Figure 3 shows change curve plots of all fifty-two experiments in the “training” set. Since the goal is to detect any change in pH, the prediction algorithm must make no distinction based on acid type. Thus, the Figure 3 plots are grouped by pH only.

Due to the clear sinusoidal patterns over the 1140–1890 MHz domain, each change curve was truncated to this domain and fed into an inverse discrete Fourier transform (IDFT) for deeper analysis. (Note: Functionally there is little difference between forward and inverse Fourier transforms; however, because the change curves were already a frequency spectrum, an inverse transform seemed more appropriate.) Because the sinusoidal curves exhibit four periods along the given domain, the inverse discrete Fourier transform of these curves shows clear impulses at coefficient 4. The magnitude of IDFT coefficients 0 through 7 for each experiment can be seen in Figure 4.

If the plots in Figure 3 depicted a true frequency domain plot of the signal received by the antenna, these inverse Fourier transforms would imply the antenna experienced a delayed voltage impulse. However, the Figure 3 plots actually depict a measure of the total energy received by the Rx antenna when the Tx antenna was transmitting a wave at the given frequency. These energy values are measured one at a time for each frequency and thus do not represent the Fourier transform of a single time-domain signal.

The value of these IDFT coefficients can be used to predict the pH of the analyte. There is high selectivity between the coefficient 4 values for pH 2, pH 3, and pH 4 (for which coefficient 4 is nearly zero). The pH 1 curves also show some difference in coefficient 4, but this is not as clear. However, because the pH 1 curves in Figure 3 appear to be shifted upwards as well, especially in the right half (1500–1890 MHz), their inverse Fourier transforms all have large values in coefficients 0, 1, and 2 as well.

On the other hand, the change curves for pH 4 and pH 5 cannot be easily distinguished from “mock” change curves, as they do not show any sinusoidal behavior or consistent vertical shifts. The pH 5 tests, in particular, are also quite misbehaved, as they contain some outliers with large vertical shifts, as seen in Figure 5.

The most severe pH 5 outlier, visible as the highest blue line in Figure 5, is suspected to have been caused by a connection issue with the SMP cables. This cable issue was not noticed until after experiments had concluded and suggests that it may be wise to switch to SMA connectors in future work. The curve’s periodic shape is actually significant enough to give it a substantial coefficient 4 value in its IDFT, placing it between that of the pH 2 and pH 3 curves (this can be seen in Figure 4). However, it can still be distinguished from all pH 2 and pH 3 curves by its substantial vertical shift, represented by its IDFT coefficient 0.

### Detection Algorithm

These results were used to build a pH change detection algorithm that could theoretically be used to detect pH changes in real time. For this limited-scope project, it was decided to use a classification method rather than a regression method, meaning that the detection algorithm would read in data and predict either “No change” or an integer pH value from 1 to 5, rather than attempting to predict each pH as a floating-point value.

It is clear that distinguishing pH values 1, 2, and 3 from pH values 4 or higher is easier than separating pH 4, pH 5, and “No change”. Thus, it was decided to split the detection algorithm into two stages. The first stage would use a pre-trained model to classify an experiment as pH 1, pH 2, pH 3, or “pH 4+”, then a separate pre-trained model would further classify the “pH 4+” experiments as pH 4, pH 5, or “No change”. The 52 experiments from the “training” set were used to train each model, while the remaining 53 experiments were fed into the final algorithm to evaluate its accuracy.

As seen in Figure 3, it takes a combination of Fourier coefficients to fully distinguish each pH value; thus, pH classification can be predicted from these coefficients using a Linear Discriminant Analysis (LDA) model. The specific software employed was the LinearDiscriminantAnalysis package from scikit-learn version 1.6.1 [15] in Python 3.11.

In the first stage, the magnitudes of Fourier coefficients 0 and 4 were used to train a two-dimensional LDA classifier with the categories “pH 1”, “pH 2”, “pH 3”, and “pH 4+”. After training the classifier on the entire “training” set, it was able to predict all of the pH classifications in the “test” set with 100% accuracy. The decision boundaries, along with the “training” and “test” data, can be seen in Figure 6, while the confusion matrix for this classifier can be seen in Figure 7.

The second classifier was designed to deal only with pH 4+ experiments, attempting to classify them as either pH 4, pH 5, or “No change”. To guard against data leakage, the same training and test sets described above were used for this second phase. As seen in Figure 5, these three categories are much more difficult to distinguish; however, there is still some noticeable variability. In particular, plotting the real parts of the IDFT coefficients rather than their magnitudes revealed some potential separability. Scatter plots comparing the real parts of coefficients 0, 1, and 2 are shown in Figure 8.

A new LDA model was trained using the real parts of the IDFT coefficients for pH 4, pH 5, and “mock” experiments from the training set. To maximize the performance and efficiency of this model, stepwise regression was employed by starting with IDFT coefficient 0 alone, then adding subsequent coefficients into the model one at a time while tracking its resulting prediction accuracy on the “test” series. After the inclusion of coefficient 4, no further improvements to the model could be made. Thus, the “second phase” of the pH detection algorithm became a five-dimensional LDA classifier, using the real parts of IDFT coefficients 0, 1, 2, 3, and 4. The confusion matrix in Figure 9 shows the performance when using this model to classify all of the pH 4+ experiments from the “test” set into pH 4, pH 5, and “No change”.

Putting these two LDA classifiers together creates a two-step pH prediction algorithm. For a given change curve, the magnitudes of IDFT coefficients 0 and 4 are first fed into the two-dimensional LDA model to classify it as pH 1, 2, 3, or 4+. For those classified as pH 4+, the real parts of coefficients 0 through 4 are then fed into the five-dimensional LDA model to further classify it as pH 4, pH 5, or “No change”. The confusion matrix for this complete algorithm applied to all 53 experiments in the “test” set is shown in Figure 10. This combined algorithm has a prediction accuracy of 81%, including 100% accuracy for pH values 1, 2, and 3.

For comparison, the same training and test sets were also used to assess a neural network-based classification algorithm, using the Multi-layer Perceptron classifier package from sci-kit-learn version 1.6.1 [15] in Python 3.11. To give this model the best chance, both the real and imaginary parts of IDFT coefficients 0 through 4 were used as the inputs, giving the input layer a size of 10. After some experimentation, the best performance was found by using the BFGS optimization algorithm and a single hidden layer of size 10. The confusion matrix for this approach can be seen in Figure 11. Although it performed remarkably well, the prediction accuracy was less than that of the LDA-based algorithm, at just 74%, and it failed to achieve 100% prediction accuracy for pH values 1, 2, and 3.

Experimenting with both LDA and neural network-based classifiers helped confirm LDA as the proper choice for this type of data.

## 4. Discussion

These results demonstrate the power of combining a sensor employing RF spectroscopy and analysis using a discrete Fourier transform to detect and measure pH changes caused by small changes to a liquid solution. The use of Fourier analysis was a particularly novel feature in this work. While previous studies have demonstrated the sensor’s ability to detect changes in blood glucose levels [14,16,17], the data processing for these studies was performed on the raw values received by the sensor. This suggests interesting analysis possibilities in related studies using Fourier transforms in the future.

When solutions were split into four categories (pH 1, 2, 3, or 4+), an LDA model trained using the Fourier coefficients achieved 100% prediction accuracy. When pH 5 was included as its own category, the model was less successful. This is unsurprising, as a change from pH 5.5 to pH 5 is quite small. Even these results were encouraging; however, 70% of pH 5 solutions were identified as pH 5, and the remaining 30% were classified as pH 4 as opposed to “No change”. Indeed, across the 48 experiments in the “test” set where pH was changed, the model successfully detected change in 47 of these, giving a positive predictive rate of 98% for pH change detection of any kind. Equally interesting is the simplicity of the final model; while many Fourier coefficients were available to the model, the predictive accuracy for the pH 1, 2, 3, and 4+ classifier was achieved using a rather simple two-dimensional LDA model, allowing for 2D graphs such as the one shown in Figure 6.

The experiment series used for this analysis is limited in scope. For example, all experiments started with pure water and ended at a lower pH. There were no experiments that involved first lowering the pH and then raising it again to confirm the response curve returns to its original position, or vice versa. Thus, it is possible that the measurements are more indicative of the solution’s electrical conductivity than its pH, and future work will attempt to distinguish these types of measurements. One way to achieve this would be by comparing the response of a particular acid and base to that of a pH-neutral salt solution formed by the same acid and base.

Other important factors to consider in regard to this technology are the ability to measure through different container materials, the ability to detect sudden variations in pH, and the ability to account for temperature fluctuations. As acknowledged in the introduction, this type of instrument would be unsuitable for containers made from RF-shielding materials, which rule out metallic pipes. For now, it seems reasonable to focus mainly on PVC pipes. Regarding sudden pH changes, the data from each experiment fed into the detection algorithm were, in fact, the average of 47 sweeps collected over 120 s, implying a two-minute minimum change detection time. However, limiting each RF sweep to the region of frequencies actually used in the algorithm (1140–1890 MHz) theoretically reduces this time to 24 s. Future experiments could also perform fewer sweeps to try to demonstrate an even shorter detection time. Finally, it is likely that temperature will affect the RF response curves. This effect will need to be characterized via another series of experiments, comparing the “change curve” shapes of analytes at various temperatures.

## 5. Conclusions

This study demonstrates the ability of a non-invasive sensor to detect pH change in a liquid solution. Despite the limitations of this study, it is a promising start, as even very small changes to the liquid solution were detected by the algorithm with 100% accuracy. To achieve pH 3, only 104 µL of water out of 200 mL total was replaced with 1 M H_2_SO_4_ (or 200 µL in the case of 1 M HCl). This demonstrates that the sensor was able to successfully detect a change in ion concentration on the order of 1 mM (0.001 mol/L) every time such a change was applied. This 1 mM value serves as the most useful metric of sensitivity for now, as the logarithmic nature of the pH scale makes it difficult to assign an overall “pH sensitivity” with the current amount of data. Nevertheless, it is reasonable to assume this sensor could detect many other types of small changes, such as the presence of hydroxide ions as a means of measuring pH values greater than 7. Future work will explore these possibilities.

## Figures and Tables

**Figure 1 sensors-25-02832-f001:**
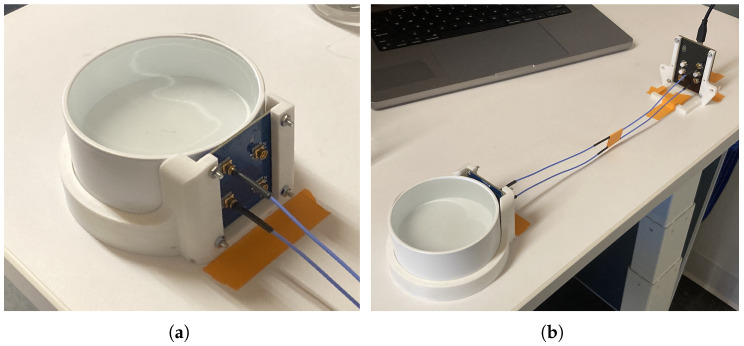
Experimental setup: (**a**) Antenna board positioned against the PVC vessel. (**b**) Connection via SMP cables to separate RF-generating board.

**Figure 2 sensors-25-02832-f002:**
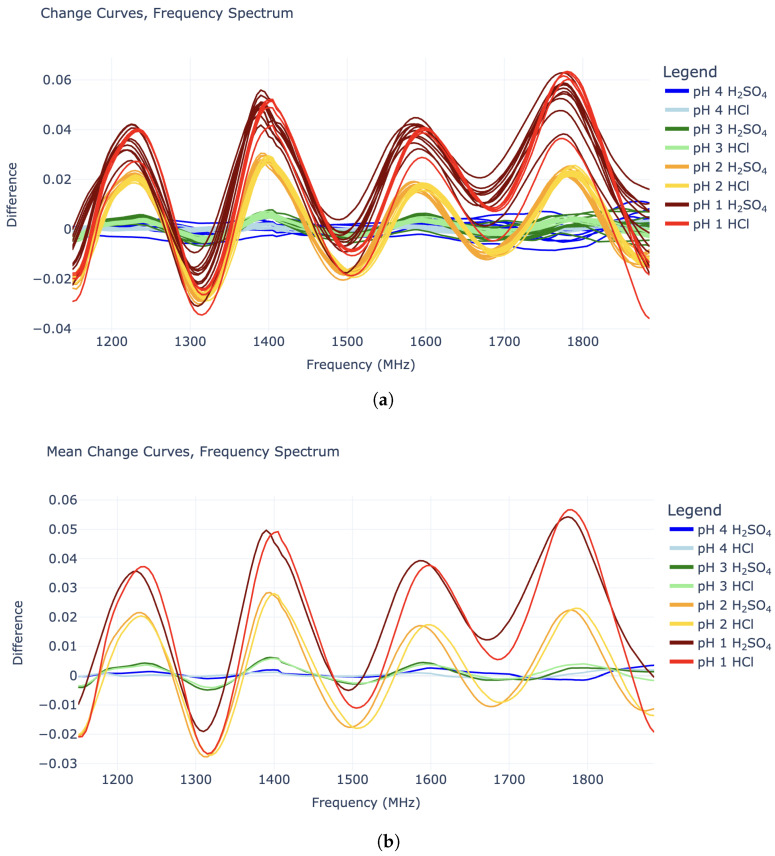
Energetic response vs. frequency plots (“change curves”) in the 1140–1890 MHz domain: (**a**) All experiment curves plotted individually. (**b**) Mean curves for each experiment type.

**Figure 3 sensors-25-02832-f003:**
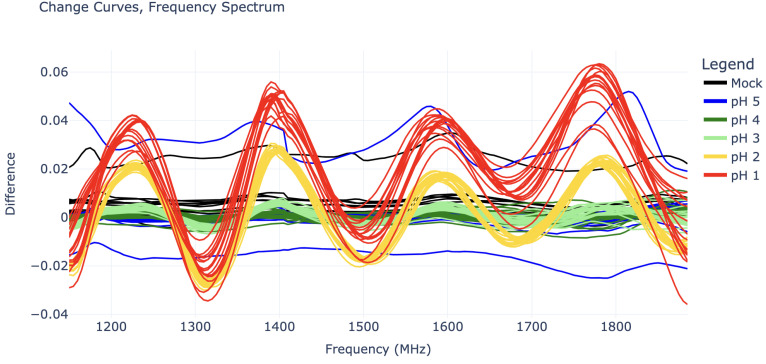
Change curves for all experiments in the “training” set.

**Figure 4 sensors-25-02832-f004:**
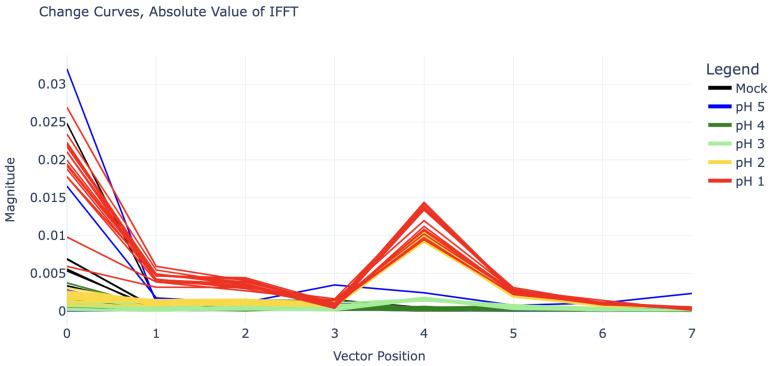
The first eight inverse discrete Fourier transform (IDFT) coefficient magnitudes for all experiments in the “training” set.

**Figure 5 sensors-25-02832-f005:**
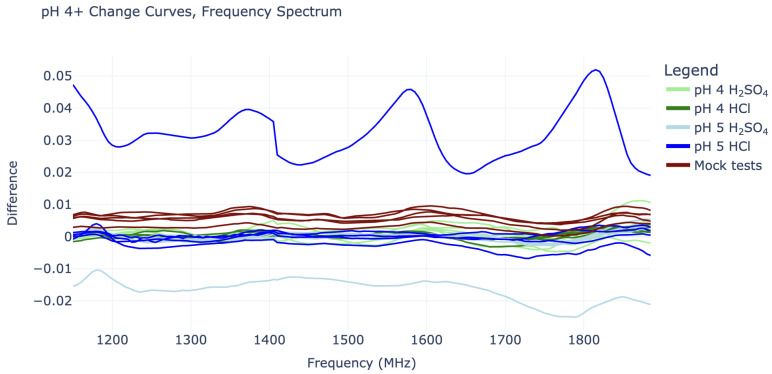
Change curves for all pH 4, pH 5, and “mock” experiments from the “training” set. Plot in frequency domain along 1140–1890 MHz.

**Figure 6 sensors-25-02832-f006:**
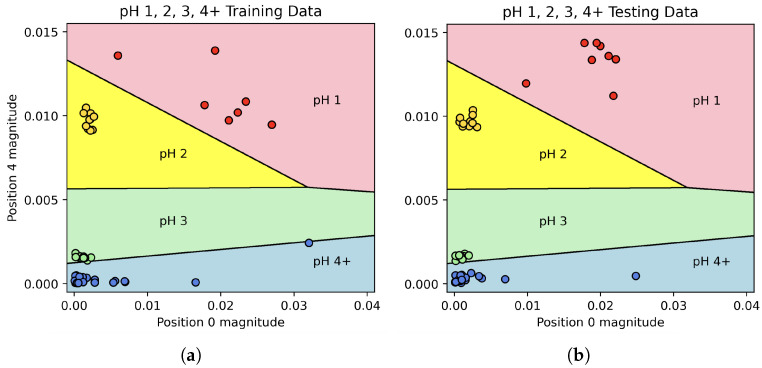
Scatter plots of IDFT vector position 0 and 4 values for (**a**) “training” data and (**b**) “test” data, along with the LDA model’s decision boundaries.

**Figure 7 sensors-25-02832-f007:**
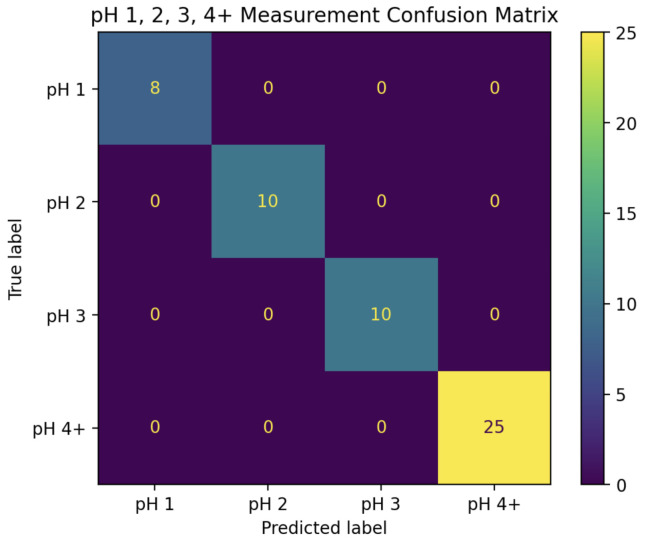
Confusion matrix for the pH 1, 2, 3, and 4+ LDA classifier applied to the “test” set, indicating its 100% accuracy.

**Figure 8 sensors-25-02832-f008:**
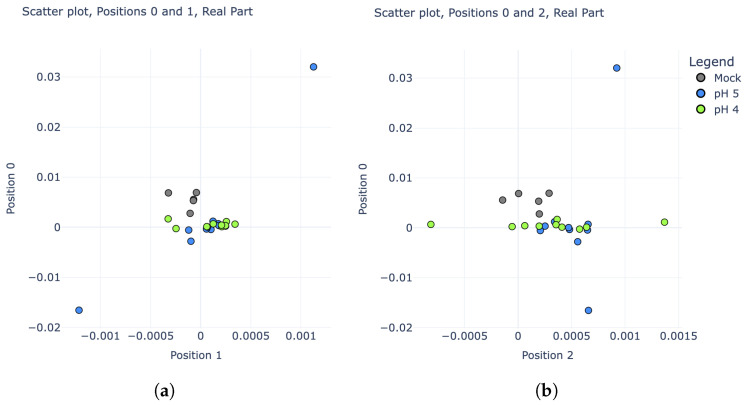
Scatter plots comparing the real parts of IDFT coefficients for pH 4, pH 5, and “mock” experiments (training set only). (**a**) Coefficients in position 0 vs. position 1. (**b**) Coefficients in position 0 vs. position 2.

**Figure 9 sensors-25-02832-f009:**
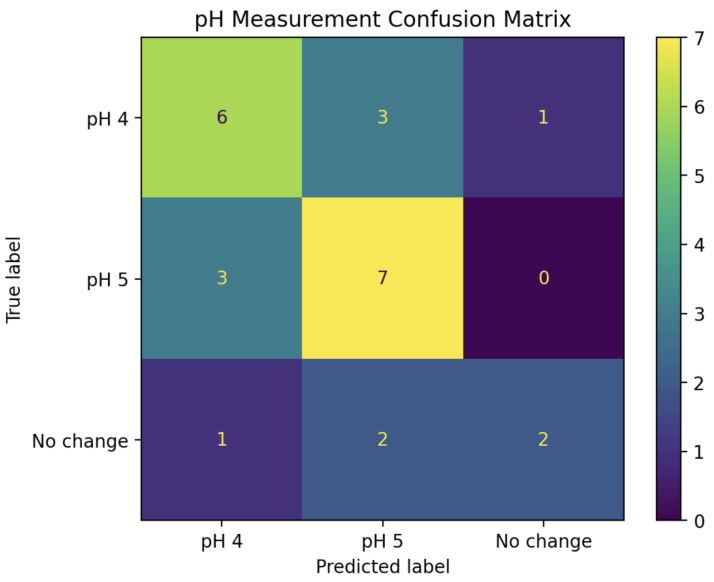
Confusion matrix for the pH 4, pH 5, and “No change” LDA classifier applied to all pH 4+ experiments from the “test” set, which by itself achieves only 60% accuracy.

**Figure 10 sensors-25-02832-f010:**
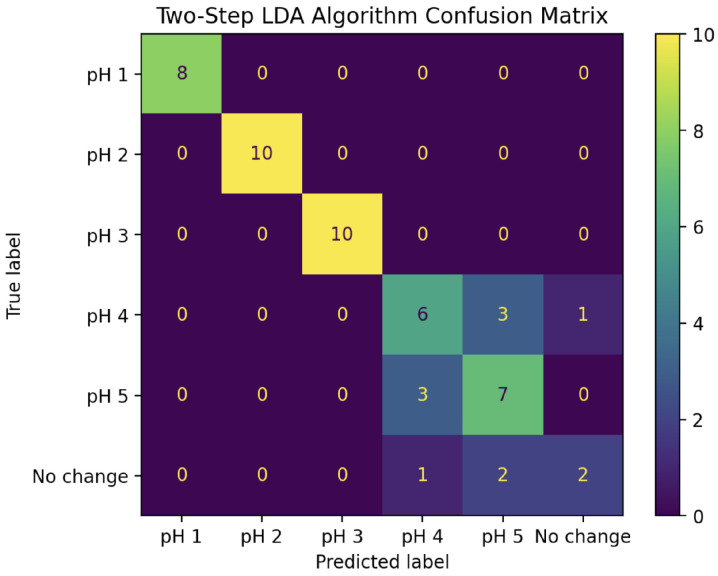
Confusion matrix for pH predictions of all 53 experiments in the “test” set using the two-step LDA algorithm. Prediction accuracy: 81%.

**Figure 11 sensors-25-02832-f011:**
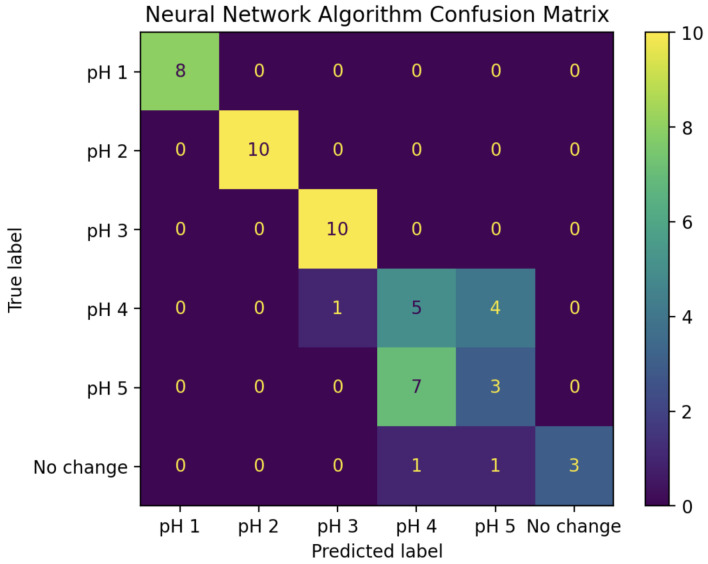
Confusion matrix for pH predictions of all 53 experiments in the “test” set using the neural network-based algorithm. Prediction accuracy: 74%.

**Table 1 sensors-25-02832-t001:** Volumes of water replaced with 1 M acid solution to create the desired final pH in the 200 mL vessel.

Final pH	1 M H_2_SO_4_ Volume (µL)	1 M HCl Volume (µL)
1	18,300	20,000
2	1300	2000
3	104	200
4	10	20
5	1	2

## Data Availability

Dataset available on request from the authors.

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
