# Peer review of "Contactless Detection of pH Change in a Liquid Analyte"

_sensors, 2025, doi:10.3390/s25092832_

Round 1
Reviewer 1 Report
Comments and Suggestions for Authors
Sensors-3541208.
In this study, the authors present an experiment in which they employ a radiofrequency sensor to measure pH variations in a liquid solution. They assert that this experiment is novel in several respects. Notably, they indicate that the sensor does not come into contact with the liquid; rather, it detects changes from the exterior of a PVC pipe. Furthermore, they state that the changes are identified using a linear discriminant analysis model, which utilizes values derived from an inverse Fourier transform of the frequency data as features. Ultimately, they claim that this represents the first application of Fourier analysis in non-contact pH measurement using radiofrequency technology.
- The authors need to enhance figures 2, 3, and 4.
- This manuscript lacks a conclusion section.
- The authors must demonstrate and substantiate the originality of their research.
- I suggest that the authors clarify the detection threshold of the sensor.
- The authors should present the sensitivity of the sensor utilized.
- Have the authors considered the stability of the sensor?
- Have they addressed the potential for sudden pH variations? Is the sensor adequately sensitive to detect these abrupt changes?
- Have the authors optimized the detection distance? It is important for them to consider the impact of losses, particularly since the sensor operates at high frequencies.
No comment
Author Response
See attached PDF

Reviewer 2 Report
Comments and Suggestions for Authors
The paper explores a novel method for contactless pH measurement using a radiofrequency (RF) sensor. Unlike conventional electrode-based pH meters that require direct contact with the analyte, this approach employs an RF-based sensing mechanism that detects pH variations externally through a PVC container. The analysis is performed using Discrete Fourier Transform (DFT) and Linear Discriminant Analysis (LDA) to classify pH values within the range of 1 to 5. The method achieves 81% overall accuracy, with 100% precision for pH values 1–3.
- The study is limited to pH values between 1 and 5, excluding neutral and alkaline solutions. Why?
- The influence of ionic species beyond HCl and Hâ‚‚SOâ‚„ on RF signals has not been discussed. Could the study be extended to include neutral and basic pH conditions to assess the sensor’s full range? Furthermore, how does the sensor respond to different acids and salts in terms of selectivity?
- The study does not assess how temperature fluctuations, solution conductivity, or the presence of additional dissolved species, such as salts and gases, may affect RF readings. Additionally, the use of a PVC container in the experiments may not fully reflect real-world scenarios where metallic elements, sediments, or turbulence could be present. How does the sensor perform under varying environmental conditions? Has it been tested in real-world industrial or biomedical applications to evaluate its practical performance?
- While LDA is effective, the paper does not discuss why it was chosen over other potential machine learning models, such as Principal Component Analysis (PCA), Neural Networks, or Wavelet Transforms. How does LDA compare to alternative data processing techniques in terms of accuracy and efficiency? What is the justification for selecting LDA over other machine learning approaches?
This study presents a novel and well-executed approach to non-invasive pH sensing, with a strong experimental methodology and effective machine learning integration. It would benefit from further validation in real-world conditions, broader pH testing, and exploration of alternative data analysis techniques. Overall, the study is well-structured and provides valuable insights into RF-based pH sensing. With minor revisions to address the points raised, the paper is suitable for acceptance
Comments on the Quality of English Language- Some sentences are overly complex and could be restructured for better readability. Minor grammatical corrections and refinements in technical descriptions would improve clarity.
Author Response
See attached PDF

Reviewer 3 Report
Comments and Suggestions for Authors
This study presents a novel approach to non-invasive pH detection using radiofrequency (RF) spectroscopy combined with inverse discrete Fourier transform (IDFT) and linear discriminant analysis (LDA). This work introduces Fourier analysis as the key to RF-based pH sensing. While the results are promising, I have the following reservation about this study, which should be addressed before publication.
- It was observed that the experiments mainly focused on reducing pH with HCl and H2SO4. To validate broader applicability, it is recommended to supplement tests that use alkalis to increase pH or other acids to regulate pH.
- It is suggested that the author give a detailed explanationof the rationale for using IDFT and the selection of the 1140-1890 MHz frequency band. How do these choices relate to the physical properties of the analyte?It is recommended to cite corresponding literature to further demonstrate the impact of the current research works such as Sensors & Actuators: B. Chemical 425 (2025) 136996; Microchemical Journal 204 (2024) 110987.
- As mentioned by the authors,the outliers of pH 5 experiment were caused by a connection issue with the SMP cables, but without specific improvement measures. Here, could the authors give a discussion in greater depth?
- It wasobserved that some figures lack sufficient resolution and axis labels, such as Figure 2-5, making trends difficult to interpret.It is suggested that the author should standardize the drawing of the whole paper.
Author Response
See attached PDF

Round 2
Reviewer 1 Report
Comments and Suggestions for Authors
My opinion this paper can be accepted for publication.
Comments on the Quality of English LanguageThis work is well corrected and I recommend it for publication.